# Dynamic Open-book Prompt for Conversational Recommender System

**Xuan Ma[1]** and **Tieyun Qian[1,2]*** and **Ke Sun[1]**

[1] School of Computer Science, Wuhan University, China

[2] Intellectual Computing Laboratory for Cultural Heritage, Wuhan University, China

{yijunma0721,qty,sunke1995}@whu.edu.cn

## Abstract

Conversational Recommender System (CRS) aims to deliver personalized recommendations through interactive dialogues. Recent advances in prompt learning have shed light on this task. However, the performance of existing methods is confined by the limited context within ongoing conversations. Moreover, these methods utilize training samples only for prompt parameter training. The constructed prompt lacks the ability to refer to the training data during inference, which exacerbates the problem of limited context. To solve this problem, we propose a novel **D**ynamic **O**pen-book **P**rompt approach, where the open book stores users' experiences in historical data, and we dynamically construct the prompt to *memorize the user's current utterance* and *selectively retrieve relevant contexts* from the open book. Specifically, we first build an item-recommendation graph from the open book and convolute on the graph to form a base prompt which contains more information besides the finite dialogue. Then, we enhance the representation learning process of the prompt by tailoring similar contexts in the graph into the prompt to meet the user's current need. This ensures the prompt provides targeted suggestions that are both informed and contextually relevant. Extensive experimental results on the ReDial dataset demonstrate the significant improvements achieved by our proposed model over the state-of-the-art methods. Our code and data are available at https://github.com/NLPWM-WHU/DOP.

## 1 Introduction

Conversational Recommender System (CRS) (Jannach et al., 2021; Gao et al., 2021; Kang et al., 2019; Zhou et al., 2021) has emerged as a promising approach for providing personalized recommendations to users through natural language utterances (Li et al., 2018a; Hayati et al., 2020). In contrast to the traditional recommender system, CRS

---
\* Corresponding author

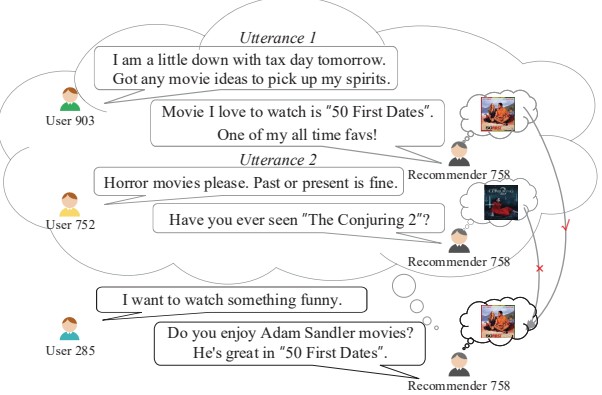

Figure 1: A real example of the "open-book" recommendation process from the ReDial dataset.

involves two primary participants, i.e., the user seeking recommendations and the recommender providing personalized recommendations based on individual experiences. The recommender performs two closely related subtasks: *the item recommendation subtask*, which predicts the item that the user might be interested in based on the utterance context, and *the response generation subtask*, which generates human-like responses in reply to the user's previous utterance.

Early methods in CRS aim to ensure semantic consistency in two subtasks by employing various word-item alignment strategies, such as mutual information maximization (Zhou et al., 2020) and contrastive learning (Zhou et al., 2022). Recent advances in pre-trained language models (PLMs) (Gao et al., 2020; Min et al., 2021; Zhang et al., 2019; Li and Liang, 2021) have introduced new insights through prompt learning. This line of work is more data-efficient and flexible. Notably, a pioneering work (Wang et al., 2022) explores the generation of the prompt by fusing item semantics from external data source and word semantics in an utterance, thereby enabling the transfer of knowledge from PLMs to CRS.

Despite the remarkable progress in prompt learning-based methods for CRS, the current

prompt design suffers from a severe challenge, i.e., the limited context within ongoing conversations especially at the inference phase. In the conventional training and testing scenario, the training data is solely used to learn prompt parameters and optimize the probabilities of ground-truth labels, with subsequent evaluation conducted on the testing data without extra information. However, it is hard for such prompt to memorize all training samples and has the risk of forgetting them, thereby exacerbating the problem of limited context. Consequently, it hampers the generalization performance in finite dialogues during inference. This limitation can be analogized to studying from a book but experiencing difficulties in recalling learned knowledge during a "closed-book" examination (Meng et al., 2021; Chen et al., 2022).

To address this limitation, it is worth considering the analogical learning process observed in human cognition. Humans have the ability to retrieve relevant skills from deep memory through associative learning, enabling them to solve problems even in situations with limited contextual information. By leveraging the past successful recommendations as an "*open-book*" knowledge store, the recommender can learn and improve her recommendation strategies through a process akin to copying from a book rather than mere memorization. Furthermore, not all knowledge in the store is relevant to current decision. A selective strategy is necessary to effectively activate memory for current recommendation. For instance, in Figure 1, the recommender 758 has successfully suggested the movie *"50 First Dates"* to the user 903 and the movie *"The Conjuring 2"* to the user 752 based on their specific needs within the given context. However, when the user 285 arrives with a preference for a funny movie, only the interaction with the user 903 can be referred to as she shares similar preference with the user 285.

Motivated by these observations, we propose a novel dynamic open-book prompt method for CRS. On one hand, we aim to empower the prompt with historical experiences to enhance its contextual awareness within finite dialogues. On the other hand, we aim to tailor the prompt to fit the target user's needs in the current utterance. Specifically, our prompt aggregates the recommender's historical accurate recommendations in the training phase via a global recommender-item graph, serving as an "open-book" knowledge store. Furthermore, we incorporate the local utterance-level semantics which memorize the current context and reflect the user's comprehensive needs. To prioritize the target user's needs when forming the prompt, we dynamically retrieve similar contexts from the open-book knowledge store by fine-tuning the prompt's representation learning process which consists of three strategies, i.e., enhancing the node embedding, enhancing the graph structure, and passing the message. In this way, our model enables the prompt to provide informed and contextually relevant suggestions, thus guiding the PLM to generate the item-aware representation. Extensive experimental results on the ReDial dataset have shown the effectiveness of our proposed model.

## 2 Related Work

**Conversational Recommender System** Conversational recommender systems have attracted attention due to recent advances in dialogue systems. Li et al. (2018a) have made substantial contributions by introducing the ReDial dataset, a basic resource in this field. To bridge the semantic gap between natural language expressions and users' preferences at the item level, researchers have explored various types of external data sources (Chen et al., 2019; Ma et al., 2020). Another research direction focuses on aligning diverse types of data (Zhou et al., 2020; Speer et al., 2017).

Previous approaches to CRS have struggled with the integration of the recommendation and conversation module due to their disparate architectures. More recently, prompt learning has emerged as a promising solution to fill this gap. For example, Wang et al. (2022) leverage the knowledge-enhanced prompt based on a fixed pre-trained language model to fulfill both subtasks in a cohesive manner. However, the prompt in Wang et al. (2022) suffers from the limited context within ongoing conversations and has suboptimal performance. To address this issue, we propose to construct a dynamic open-book prompt-based model which explicitly creates an open-book knowledge store and retrieves relevant contexts as references from it, which can enhance the prompt's generalization capabilities during inference.

**Graph Learning** Graph convolutional networks have found extensive applications across various tasks (Wu et al., 2020) by leveraging their ability to merge vertex features and input graph topology for efficient node embedding learning (Kipf and Welling, 2016). However, the construction of the

graph structure, typically performed manually or using the k-nearest neighbors method, is pretty susceptible to noise. Hence researchers propose to learn adaptive graphs for facilitating downstream graph-based tasks (Veličković et al., 2017; Li et al., 2018b; Fatemi et al., 2021). However, existing CRS methods still adopt a fixed graph which is unable to adapt to different conversations. To overcome this limitation, we propose to fine-tune the item-recommendation graph by enhancing the recommender embedding, the graph structure, and the message passing process when constructing the prompt. Our enhancement allows for dynamic retrieval of pertinent information from a static open-book knowledge store, satisfying diverse users' needs and evolving conversations.

**Prompt Learning** Prompt learning is a technique that can enhance performance in various tasks by incorporating additional information, known as a prompt, into the generation process of pre-trained language models (PLMs) (Lester et al., 2021). Typically, the prompt is inserted at the beginning of the PLM's input to exert better control over its generation (Gao et al., 2020; Radford et al., 2021; Jin et al., 2021). Initially, the manual design of discrete prompts is a mainstream (Raffel et al., 2020; Brown et al., 2020; Petroni et al., 2019; Hsu et al., 2022). However, such prompts lack a fine-tuning process and require extensive experiments, experience, and expertise in language, incurring significant costs. Recent studies have paid more attention on automatically optimizing continuous prompts (Gu et al., 2021; Hu et al., 2021; Lester et al., 2021; Liu et al., 2022). We are also in line with this trend.

## 3 Methodology

In this section, we present our proposed **D**ynamic **O**pen-book **P**rompt (DOP) approach for CRS. We start by introducing the basic open-book prompt, which leverages users' experiences from training samples to enrich the current context and facilitate decision-making even during inference. Next, we design three strategies to incorporate utterance-level semantics, which allow for dynamic retrieval in the "open-book" knowledge store to meet the specific needs of the target user. Following this, we finally describe how the prompt guides the PLM when performing the CRS subtasks. Figure 2 (a) illustrates a typical prompt learning framework for CRS.

### 3.1 Basic Open-book Prompt

The recommender plays a crucial role in two key subtasks: providing recommendations and responding to the previous utterance. In the item recommendation subtask, the recommender's perception significantly influences the recommended items. Additionally, ensuring the coherence and consistency in dialogues is paramount for the response generation subtask.

To achieve these, we first leverage the recommender's representation as the foundational prompt to guarantee the coherence in recommendation behavior and response style for both subtasks.

We then leverage successful historical recommendations to build a recommender-item graph $G = \{S \cup V, \boldsymbol{R}\}$, where $S$ represents the vertex set of $|S|$ recommender nodes, denoted as $\{s_1, s_2, ..., s_{|S|}\}$, and $V$ represents the vertex set of $|V|$ item nodes, denoted as $\{v_1, v_2, ..., v_{|V|}\}$. The binary matrix $\boldsymbol{R} \in \mathbb{R}^{|S| \times |V|}$ describes the recommender-item interactions, where $R_{sv} = 1$ if the recommender $s$ has correctly recommended the item $v$ and $R_{sv} = 0$ otherwise. The graph $G$ will serve as a graph-based reference book in an open-book exam. During testing, the knowledge can be directly extracted from this "book" and used for accessing memory and enhancing the final recommendation decisions.

Finally, we employ graph convolution to get the recommender's representation which forms a basic open-book prompt. Through the integration of successful experiences stored in the open-book knowledge, the prompt gains the ability to directly extract contexts from the entire training data as the reference. This enhances the prompt's understanding capability and enables it to deliver more informed recommendations to the target user, and consequently improves recommendation performance in finite dialogues during inference.

### 3.2 Dynamic Open-book Prompt

The static "open-book" knowledge store for the prompt consists of past successful recommendations catering to various users' needs. However, it may also contain irrelevant noises, which could potentially affect the performance when handling the target user's utterance. To address this issue, we propose three strategies to integrate utterance-level semantics on the graph-based open-book knowledge store. By leveraging the utterance-level semantics that memorize the current context and cap-

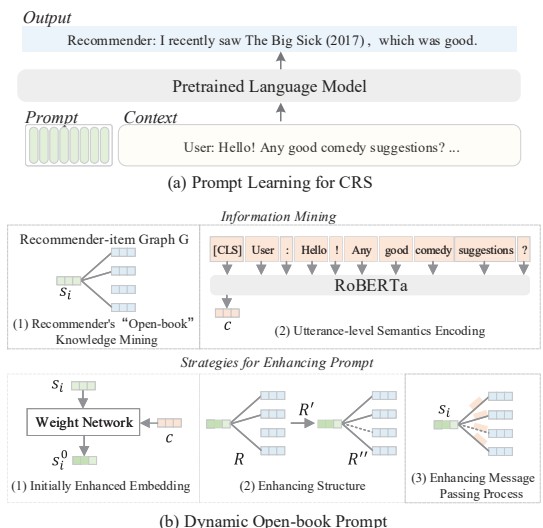

Figure 2: The overall framework of our proposed model. Gray shapes indicate that the model parameters are fixed.

ture the user's comprehensive needs, we selectively retrieve relevant contexts from the open book while dynamically filtering out irrelevant information.

The framework of our DOP is illustrated in Figure 2 (b). We now proceed to elaborate on each step in detail.

### 3.2.1 Utterance-level Semantics Encoding

We generate the utterance-level representation $c$ by first feeding the utterance into the fixed pretrained language model RoBERTa (Liu et al., 2019) and then encoding the output [CLS] vector of RoBERTa with a MLP layer.

### 3.2.2 Strategies for Enhancing Prompt

As mentioned earlier, we employ a global recommender-item graph as the static "open-book" knowledge store for the prompt, and we wish to introduce utterance-level semantics for targeted recommendations. To achieve this, we propose three strategies, i.e., enhancing the recommender's embedding, enhancing the graph structure, and passing the message during the representation learning process of the prompt. These strategies guide the prompt's targeting process and enhance the informativeness of the static "open-book" prompt.

**Initially Enhanced Embedding for the Prompt** By enhancing the quality of the recommender's embedding, we aim to capture and comprehend the semantic features expressed in the user's utterance. This augmentation results in an improved representation of the prompt, facilitating a deep understanding of the user's needs. Specifically, we introduce

a weight network that takes the utterance-level semantic representation $c$ as input and generates a channel weight $\boldsymbol{W}$ to adapt the recommender's embedding. This weight network dynamically adjusts the importance of different channels in the embedding process based on the specific characteristics of the utterance, formally described as:

$$\boldsymbol{W} = Reshape(\boldsymbol{W}_2(ReLU(\boldsymbol{W}_1\boldsymbol{c} + \boldsymbol{b}_1)) + \boldsymbol{b}_2), \quad (1)$$

where $\boldsymbol{W}_1 \in \mathbb{R}^{k \times d}$ ($k$ is the dimensionality for the hidden layer), $\boldsymbol{W}_2 \in \mathbb{R}^{d^2 \times k}$, $\boldsymbol{b}_1 \in \mathbb{R}^k$, and $\boldsymbol{b}_2 \in \mathbb{R}^{d^2}$ are weight matrices and biases. The operation $Reshape$ refers to the dimension change: $\mathbb{R}^{d^2} \to \mathbb{R}^{d \times d}$. The enhanced recommender's embedding $\boldsymbol{s}^0$ is obtained by multiplying the utterance-level semantic channel weight with the original embedding as: $\boldsymbol{s}^0 = \boldsymbol{W}\boldsymbol{s}$.

**Enhancing Structure to Form the Prompt** By enhancing the graph structure of the global recommender-item graph $G$ and adjusting the aggregating weights of items, we can prioritize items that are contextually relevant to the user's utterance during the prompt's representation learning process. This enables us to address target user's needs more effectively and improve the overall recommendation quality. Specifically, for an utterance generated by the current recommender $s_i$ and an anonymous user, we calculate a new recommender-item similarity matrix $\boldsymbol{R}' \in \mathbb{R}^{|S| \times |V|}$. In this matrix, the entries in the $i$-th row represent the affinity based on the current utterance between the recommender $s_i$ and all items in $V$, while the entries in other rows are set to zero. To calculate each entry in the $i$-th row, we employ the following similarity measurement:

$$\boldsymbol{R}'_{ij} = \sigma(\frac{< \boldsymbol{s}_i^0 \times \boldsymbol{W}_3, \boldsymbol{v}_j \times \boldsymbol{W}_4 >}{|\boldsymbol{s}_i^0 \times \boldsymbol{W}_3||\boldsymbol{v}_j \times \boldsymbol{W}_4|}), j \in \{1, ..., |V|\},$$
(2)

where $\sigma$ is the sigmoid function, $\boldsymbol{W}_3$ and $\boldsymbol{W}_4$ are two trainable matrices, $<, >$ denotes vector inner product operation, $\boldsymbol{s}_i^0$ denotes the enhanced embedding for the recommender $i$, and $\boldsymbol{v}_j$ denotes the embedding for the item $j$.

Besides, we only retain the entries with the top-$k$ computed similarities in the $i$-th row to save the computational cost. Next, we combine the generated matrix $\boldsymbol{R}'$ with the original matrix $\boldsymbol{R}$ in $G$, resulting in an enhanced graph structure $\boldsymbol{R}''$. This new matrix incorporates the contextual information from the current utterance and the recommender $s_i$, resulting in a targeted recommender-item interaction matrix. We also introduce a constraint

loss term denoted as $L_{con}$, defined as the Frobenius norm between $\boldsymbol{R}'$ and $\boldsymbol{R}$, i.e., $L_{con} = ||\boldsymbol{R}' - \boldsymbol{R}||_F^2$, ensuring the preservation of the original graph structure during this procedure.

**Enhancing Message Passing Process to Form the Prompt**   By enhancing the message passing process, we seamlessly incorporate the comprehensive user's preferences, including likes and dislikes expressed in the current utterance. This enhancement empowers the prompt to directly capture the associations between utterances and items, enabling it to adjust the understanding of user's preferences and aggregate neighborhood messages accordingly. With the enhanced graph structure $\boldsymbol{R}''$, recommender node embeddings and item node embeddings, we propose an utterance-aware graph convolutional layer. Additionally, we introduce a filter component that interacts with contextual semantic features. This component plays a crucial role in controlling the information flow during the message passing process, allowing for flexible re-weighting of the impacts between utterances and items when needed.

Specifically, given the enhanced recommender's representation $\boldsymbol{s}_i^0$ and the item's representation $\boldsymbol{v}_j$ ($\boldsymbol{v}_j = \boldsymbol{v}_j^0$), the output of graph convolutional operation at the $l$-th layer is as follows,

$$\boldsymbol{s}_i^l = \sum_{v_j \in \mathcal{N}(s_i)} \frac{\boldsymbol{v}_j^{l-1} + PReLU(\boldsymbol{W}_1^c \boldsymbol{c}_i + \boldsymbol{b}_1^c)(\boldsymbol{c}_i \odot \boldsymbol{v}_j^{l-1})}{\sqrt{\sum_{k=1}^{|V|} \boldsymbol{R}_{ik}''} \sqrt{\sum_{k=1}^{|S|} \boldsymbol{R}_{kj}''}}, \quad (3)$$

$$\boldsymbol{v}_j^l = \sum_{s_i \in \mathcal{N}(v_j)} \frac{\boldsymbol{s}_i^{l-1} + PReLU(\boldsymbol{W}_1^c \boldsymbol{c}_i + \boldsymbol{b}_1^c)(\boldsymbol{c}_i \odot \boldsymbol{v}_j^{l-1})}{\sqrt{\sum_{k=1}^{|V|} \boldsymbol{R}_{ik}''} \sqrt{\sum_{k=1}^{|S|} \boldsymbol{R}_{kj}''}}, \quad (4)$$

where $\mathcal{N}(s_i)$ and $\mathcal{N}(v_j)$ denote the neighbors of the recommender $s_i$ and item $v_j$ in the enhanced graph structure, and $\boldsymbol{c}_i$ denotes the utterance's embedding between the recommender $s_i$ and the target user during the conversation.

**Finally Enhanced Embedding as the Prompt**   After the propagation process in the utterance-aware graph convolutional layer, we combine the outputs obtained at each layer to form the representation of the recommender $s_i$ and item $v_j$: $\boldsymbol{s}_i = \sum_{l=0}^L \boldsymbol{s}_i^l / L; \boldsymbol{v}_j = \sum_{l=0}^L \boldsymbol{v}_j^l / L$, where $\boldsymbol{s}_i$ is the prompt and $\boldsymbol{v}_j$ is the item's representation.

## 3.3   Training

Inspired by PREFIX-TUNING (Li and Liang, 2021), we leverage the final representation of the recommender $\boldsymbol{s}_i$ in the utterance as a general prefix

to adapt the PLM for both CRS subtasks. Formally, we prepend the prefix to the input utterance sequence $\mathcal{X}$ as follows,

$$\mathcal{X}' = [\boldsymbol{s}_i; \mathcal{X}]. \quad (5)$$

Subsequently, we provide the entire input $\mathcal{X}'$ to the PLM and calculate the decoder matrix in the last layer to facilitate learning for both the item recommendation subtask and the response generation subtask.

**Item Recommendation Subtask**   From the decoder matrix, we extract the vector $\boldsymbol{x}$ for the last token in $\mathcal{X}'$ as the user's representation in the current utterance. Then, we compute the probability to recommend an item $\boldsymbol{v}_j$ for the user in the current utterance as:

$$P(\boldsymbol{x}\boldsymbol{v}_j) = \text{softmax}(\boldsymbol{x}^T \boldsymbol{v}_j). \quad (6)$$

We use the sum of the cross-entropy of each item's prediction and the ground truth as the recommendation loss for the current utterance:

$$L_{rec} = -\sum_{j=1}^{|V|} [y_j \cdot \log P(\boldsymbol{x}\boldsymbol{v}_j) + (1-y_j) \cdot \log(1 - P(\boldsymbol{x}\boldsymbol{v}_j))], \quad (7)$$

where $y_j$ denotes a binary ground-truth label which is equal to 1 when the item $v_j$ is recommended for the user in the current utterance.

Finally, considering both the recommendation loss and the constraint loss, we define a joint loss function for the recommendation subtask as follows:

$$L_r = L_{rec} + \alpha L_{con}, \quad (8)$$

where $\alpha$ is the hyperparameter that balances the two loss terms.

**Response Generation Subtask**   Following the conventional setting of the response generation task, we employ the standard maximum likelihood estimator (MLE) as our optimization objective:

$$L_{gen} = -\sum_{k=1}^L \log P(w_k | \boldsymbol{s}_i; w_{<k}), \quad (9)$$

where $L$ is the length of the current utterance, $w_k$ is the $k$-th word in the utterance, $w_{<k}$ denotes the words proceeding the $k$-th position and $P(\cdot)$ denotes the distribution over the vocabulary for the next $k$-th word in the current utterance. Finally, the joint loss function combining $L_{gen}$ and $L_{con}$ for the response generation subtask is:

$$L_g = L_{gen} + \alpha L_{con}. \quad (10)$$

Note two subtasks in CRS are conducted separately, and we use the same hyperparameter $\alpha$ to reduce the overhead.

## 4 Experiments

This section presents experimental results and the analysis.

### 4.1 Experimental Setup

*Dataset* We conduct experiments on ReDial (Li et al., 2018a), a real-world CRS dataset dedicated to movie recommendation. It consists of 10,006 conversations, with a total of 182,150 utterances involving 51,699 movies. We split the data into training, validation, and test sets using an 8:1:1 ratio.

*Settings* We implement models in PyTorch and train them on a NVIDIA 3090. The base PLM we use is DialoGPT-small (Zhang et al., 2019), which is pre-trained on 147M dialogues collected from Reddit, following UniCRS. Throughout the training process, the parameters of the base model remain fixed. The item embedding size and recommender embedding size are both set to 768, consistent with the input embedding size of DialoGPT-small. We employ 2 graph convolutional layers ($L = 2$). During training, the batch size for the item recommendation subtask is 32, and for the response generation subtask, it is 8. For optimization, we employ the Adam optimizer (Kingma and Ba, 2014) with default parameter settings. The learning rate is set to 0.0005 for the item recommendation subtask and 0.0001 for the response generation subtask. The values of parameters $k$ and $\alpha$ are determined based on their optimal performance on the validation set, with $k$ set to 15 and $\alpha$ set to 6.

*Baselines* We compare our DOP with four groups of CRS approaches. (1) GPT-2 (Radford et al., 2019), DialoGPT (Zhang et al., 2019), and BART (Lewis et al., 2019) are pre-trained language models recognized for their effectiveness in language modeling and dialogue generation. (2) ReDial (Li et al., 2018a) is a standard reference for evaluating the performance of our task. (3) KBRD (Chen et al., 2019), KGSF (Zhou et al., 2020), and C$^2$-CRS (Zhou et al., 2022) are methods that incorporate various external data sources and employ dedicated word-item alignment strategies to bridge the semantic gap between the recommendation module and the conversation module. (4) UniCRS and BOP are prompt learning-based methods. The former is proposed by Wang et al. (2022) with external DBpedia data and the latter is our basic open-book prompt in Section 3.1.

*Evaluation Metrics* Consistent with prior re-

search in CRS, we evaluate the performance of all models on recommendation and conversation subtasks (Zhou et al., 2020; Wang et al., 2022; Zhou et al., 2020). For the item recommendation subtask, we adopt Recall@$k$ (R@$k$) as our evaluation metric, where $k$ takes on values of 1, 10, and 50. This metric measures the presence of ground-truth recommendations among the top-$k$ items generated by the models. For the response generation subtask, we use the distinct $n$-gram metric with $n$ values of 2, 3, and 4. This criterion assesses the response diversity which is a critical aspect for creating engaging and natural dialogue systems.

### 4.2 Main Results

Table 1 presents a comprehensive summary of the performance for the item recommendation subtask and the response generation subtask. Notably, our DOP model consistently outperforms other models across all metrics, demonstrating its superiority. Based on these results, we draw the following observations.

Table 1: The overall performance comparison on the ReDial dataset. The best performance among all is in bold while the second best one is underlined. Numbers marked with an asterisk (*) signify statistically significant improvements compared to the best baseline (t-test with a p-value < 0.01).

| Model | Recommendation | | | Conversation | | |
|---|---|---|---|---|---|---|
| | R@1 | R@10 | R@50 | Dist-2 | Dist-3 | Dist-4 |
| GPT-2 | 0.023 | 0.147 | 0.327 | 0.354 | 0.486 | 0.441 |
| DialoGPT | 0.030 | 0.173 | 0.361 | 0.476 | 0.559 | 0.486 |
| BART | 0.034 | 0.174 | 0.377 | 0.376 | 0.490 | 0.435 |
| ReDial | 0.023 | 0.129 | 0.287 | 0.225 | 0.236 | 0.235 |
| KBRD | 0.035 | 0.178 | 0.344 | 0.281 | 0.385 | 0.454 |
| KGSF | 0.035 | 0.179 | 0.369 | 0.311 | 0.443 | 0.545 |
| C$^2$-CRS | 0.053 | 0.232 | 0.458 | 0.387 | 0.707 | 0.874 |
| UniCRS | 0.051 | 0.238 | 0.465 | 0.437 | 0.689 | 0.880 |
| BOP | 0.052 | 0.237 | 0.460 | 0.773 | 1.212 | 1.531 |
| **DOP** | **0.057**$^*$ | **0.257**$^*$ | **0.474**$^*$ | **0.876**$^*$ | **1.428**$^*$ | **1.827**$^*$ |

(1) The approaches that leverage more external data tend to achieve superior performance compared to other baselines. By incorporating additional external data sources, approaches such as KBRD, KGSF, and C$^2$-CRS enhance the semantic understanding of words and items mentioned in the dialogue, leading to a more accurate capture of user's preferences. For instance, KGSF leverages ConceptNet and DBpedia to enhance both word-level and item-level semantic information,

surpassing KBRD, which solely incorporates the DBpedia data source.

(2) A well-designed word-item fusion strategy effectively narrows the semantic gap between words and items, resulting in a more precise performance. For example, $C^2$-CRS achieves a relatively high Recall@1 score by employing a coarse-to-fine semantic fusion strategy, which combines different semantic spaces in a multi-grained manner.

(3) The prompt-based method UniCRS shows an overall better performance than many other baselines. However, its prompt is enhanced by external data, which is not tailored for the user's need. In contrast, our DOP approach focuses on user's demands. We employ utterance-level semantics to enhance the prompt's learning process and retrieve relevant contexts in the graph-based open-book knowledge store. This enables a direct access to historical information and enhances the decision quality. In addition, the performance comparison between BOP and DOP underscores the effectiveness of dynamic strategies in integrating the utterance-level semantics. For the response generation subtask, our DOP shows significant improvements in terms of diversity. This can be attributed to the inclusion of the recommenders' representation, which encompasses their individual speaking styles and word choices. As a result, our model generates dialogues with each recommender's unique characteristics and augments the personalization beyond the scope of dialogue context.

## 4.3 Ablation Study

The core merit of our DOP lies in its dynamic prompt customization strategy, which utilizes three techniques to adjust the prompt within the graph-based "open-book" knowledge store based on semantics. We perform a series of ablation studies on each component of DOP to examine their impacts on the item recommendation subtask, including: (1) $DOP_{w/o-EM}$ removes the recommender embedding enhancement component; (2) $DOP_{w/o-LG}$ removes the graph structure enhancement component; (3) $DOP_{w/o-MP}$ removes the utterance-item interaction in the message passing process; (4) $BOP_{w/o-O}$ further removes the open-book knowledge from the BOP.

As shown in Table 2, all variants suffer from a performance decrease. We can find that each component indeed contributes to the overall performance, and combining them together makes the

Table 2: Results for ablation study on the item recommendation subtask.

| Model | R@1 | R@10 | R@50 |
|---|---|---|---|
| DOP | **0.057** | **0.257** | **0.474** |
| $DOP_{w/o-EM}$ | 0.050 | 0.243 | 0.456 |
| $DOP_{w/o-LG}$ | 0.056 | 0.256 | 0.464 |
| $DOP_{w/o-MP}$ | 0.052 | 0.247 | 0.473 |
| BOP | 0.052 | 0.237 | 0.460 |
| $BOP_{w/o-O}$ | 0.048 | 0.218 | 0.428 |

best recommendation. (1) By removing the recommender's embedding enhancement component, $DOP_{w/o-EM}$ experiences the largest performance drop since the recommender's embedding functions as a final prompt to steer the PLM, which introduces the utterance-level semantics for effectively capturing the user's need. (2) The Recall@50 score for $DOP_{w/o-LG}$ decreases most, highlighting the significance of the enhanced graph structure in effectively emphasizing relevant items based on the utterance context within the prompt. However, due to the expanded candidate pool in the prompt, achieving precision in specific recommendations like Recall@1 becomes more challenging. (3) The simple graph convolutional layer setting $DOP_{w/o-MP}$ without the utterance-item interaction is inferior to the complete DOP. This implies that the utterance-item interaction can re-weight recommender-item correlations and successfully capture the user's preference. (4) Comparing the results between $BOP_{w/o-O}$ and BOP, we observe that incorporating the open-book knowledge significantly improves the prompt. This integration leverages the training set knowledge not only for prompt parameterization but also for direct reference during testing, thus facilitating the recall of relevant information.

## 4.4 Case Study

Table 3 presents two cases of the responses generated by UniCRS and our DOP. We can observe that our DOP model outperforms the UniCRS baseline in generating more informative responses.

In Case 1, we evaluate the model's ability to generate diverse initial recommender's sentences without context. Our model successfully generates responses by leveraging the recommender's historical conversation information within the recommender representation as the base prompt. In contrast, UniCRS's prompt, which needs the ongoing utterance context, is unavailable and thus ineffective. Consequently, our model is capable of generating responses that reflect the personal

Table 3: Generation cases on the ReDial dataset.

| | |
|---|---|
| Case 1 | *Label:*
**Recommender:** Hi. I heard you were interested in finding some good movies. What kind of movies do you like?
*Generation:*
**UniCRS:** Hi!
**Ours:** Hi. I heard you might be looking for a movie. What type of movies do you like? |
| Case 2 | *Context:*
**User:** Hello! I want to go see a movie in a theater with my elderly grandma tomorrow. One she already saw and liked was A Wrinkle in Time.
**Recommender:** Oh wow okay.
**User:** She likes drama and anything British that isn't too racy.
**Recommender:** I think you need to see Ready Player One, Peter Rabbit, and Coco! Those would be great for you and her.
**User:** She might like Coco, but I think Peter Rabbit is a bit too far out there for her.
*Label:*
**Recommender:** Sounds good! I hope I helped.
*Generation:*
**UniCRS:** It is a good one.
**Ours:** Perfect, I think she would like it. |

language habits of the recommender, as opposed to the generic "*hi*" produced by UniCRS.

In Case 2, we verify the relevance of the dialogues generated by our DOP model to the preceding context. By leveraging utterance-level semantics as prompt's guidance and selecting consistent knowledge from the open book, our model captures the user's specific needs and accurately identifies the role of "grandma", resulting in comprehensive and contextually relevant responses. On the contrary, UniCRS solely relies on external knowledge to enhance the prompt, overlooking the valuable information embedded in the utterance, thus failing to consider the mentioned role within the context.

In summary, our results prove that the diversity of generated dialogues is critical for the integration of personalization and contextual information. UniCRS fails to leverage the relevant information from the recommender and lacks effective utilization of utterance-level dialogue context, leading to generic and repetitive responses. In contrast, our DOP not only integrates the recommender's representation but also employs dynamic strategies to explore the dialogue context. Thus, the generated responses are more targeted and diverse in nature.

## 4.5 Parameter Study

There are two key hyperparameters in our method: $k$ for the number of added edges and $\alpha$ to control the weight in the loss function. We only show the results of parameter study on the item recommendation subtask due to the space limitation. In Figure 3, we observe that the recall@1 score and recall@10 score exhibit a similar trend. Regarding the number of candidate edges $k$, the performance initially improves as $k$ increases from 5 to 15. However, it quickly deteriorates when $k$ becomes larger. This suggests that suitable recommender-item correlations in the prompt can effectively emphasize user's preferences, but too much information introduces noise. Similarly, the choice of the regularization coefficient $\alpha$ is crucial for optimizing the overall objective. Excessive regularization hampers performance. In our experiments, we set $k$ to 15 and $\alpha$ to 6 based on our analysis.

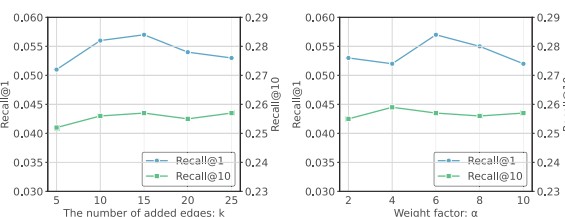

Figure 3: Impacts of hyperparameters $k$, $\alpha$ on the item recommendation subtask.

## 5 Conclusion

In this work, we propose an innovative approach to create a dynamic "open-book" prompt for the CRS task. Specifically, the open book stores historical accurate recommendations as the users' experiences, and the dynamically constructed prompt memorizes the user's current utterance and selectively retrieves relevant contexts from the open book. Our objectives are twofold. Firstly, we aim to leverage the recommender's successful experiences to enrich the context besides the user's finite dialogue. This enables the prompt to refer to the existing knowledge for decision-making during inference. Secondly, we aim to customize the prompt to match the target user's needs. Through three strategies to refine the representation learning process, our prompt can provide targeted suggestions that are both informed and contextually relevant. Extensive experiments conducted on the ReDial dataset demonstrate the superiority of our proposed model over the state-of-the-art baselines.

## Limitations

We identify two primary limitations of this study that might be addressed in future research. The first limitation pertains to the model's inability to handle cold-start scenarios where the recommender lacks historical recommendations. In such cases, the recommender's representation lacks sufficient supervision, making it unsuitable as a base prompt. To address this, we consider leveraging the demand type from dialogue semantics as the base prompt to mitigate cold-start issues in the future. The second limitation is about the appropriate utilization of external knowledge. Although our model has achieved promising results, it does not leverage any external knowledge base. The judicious incorporation of external knowledge to supplement semantic information is beneficial for natural language understanding. In future work, we plan to enhance the comprehension of dialogue content by leveraging external knowledge repositories, such as the ConceptNet.

## Ethics Statement

In this work, all authors acknowledge and adhere to the *ACL Code Ethics*, ensuring compliance with the code of conduct. This paper presents a method for constructing an open-book prompt that dynamically integrates historical recommendations with the current user's demand for conversational recommender system. For evaluation, we utilize the ReDial dataset, which consists of annotated dialogues where people recommend movies to one another. The primary language is English. Our model neither introduces social/ethical bias nor amplifies any bias present in the data.

## Acknowledgments

This work was supported by the grant from the National Natural Science Foundation of China (NSFC) project (No. 62276193). It was also supported by the Joint Laboratory on Credit Science and Technology of CSCI-Wuhan University.

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
