# OpenReview forum: "Dynamic Open-book Prompt for Conversational Recommender System"
_EMNLP/2023/Conference — EMNLP 2023 Findings_

### Official Review · Reviewer_MJfa · 2023-08-04

**Soundness:** 3

**Excitement:**

3: Ambivalent: It has merits (e.g., it reports state-of-the-art results, the idea is nice), but there are key weaknesses (e.g., it describes incremental work), and it can significantly benefit from another round of revision. However, I won't object to accepting it if my co-reviewers champion it.

**Missing References:**

No particular reference missing.

**Paper Topic And Main Contributions:**

The paper focuses on the conversational recommendation task, proposing methods to retrieve and leverage at inference time past recommender runs that are relevant to an incoming user utterance. Authors are strongly inspired by UniCRS (Wang et al., 2022), with the key differentiator being the use of past recommender runs (or "past experiences") as a "dynamic open-book." To this end, authors propose three methods for fusing the retrieved past experience into the prompt representation learning process: enhancing the node embeddings from the recommender-item graph by fusing it with the current user utterance; enhancing the recommender-item graph structure by also fusing them with the current user utterance; enhancing the message-passing step by, again, fusing it with the current user utterance. A system based on DialoGPT (similar to Wang et al., 2022) and with the three methods is tested on the ReDial dataset, as well as ablations of this system with and without each method. Results generally favor the proposed system over UniCRS, with ~10% performance gains on topmost Recall@k.

**Questions For The Authors:**

In terms of the architecture, is there any use for the dialogue history in addition to the current user utterance?

**Reasons To Accept:**

1. Conversational recommendation is an important task which has attracted increasing attention in the NLP community, particularly for the use of LLMs towards it.

2. Authors test their methods on ReDial, a well-known dataset in this space with multiple baselines available and included in the paper.

3. The first part of the results, measured in terms of Recall@k, is promising. While BOP approaches UniCRS, DOP is 11% better on Recall@1 and 8% better on Recall@50.

**Reasons To Reject:**

1. The paper lacks clarity in multiple parts. The architecture is presented in a very fragmented way (see Presentation Improvements for more details), and concepts are ill-defined. Importantly, it is never perfectly clear what exactly the "recommender nodes" are --- illustrating a single node and the actual graph would be helpful, even as an Appendix. Consequently, authors use expressions like "this enables direct access to historical information" (lines 539-540), and readers can only assume that "historical information" is defined as past recommender runs --- even though there could be multiple interpretations to "historical information."

2. The second part of the results, measured in terms of Dist-2, Dist-3, and Dist-4, is not particularly convincing. Given the more dynamic nature of prompt learning, there will be higher prompt variation; therefore, it is highly expected that there will be greater natural language variation at generation. For this reason, Dist-n should not be the measure to differentiate between methods. Instead, the evaluation question should be: How helpful is this increased natural language variation? Authors hint on this question with two qualitative examples on Table 2, but aspects like helpfulness/informativeness should be more quantitatively measured --- for example, by using human judges on a certain sample of responses.

**Reproducibility:**

3: Could reproduce the results with some difficulty. The settings of parameters are underspecified or subjectively determined; the training/evaluation data are not widely available.

**Reviewer Confidence:**

4: Quite sure. I tried to check the important points carefully. It's unlikely, though conceivable, that I missed something that should affect my ratings.

**Typos Grammar Style And Presentation Improvements:**

Unlike Wang et. al (2022), it is unclear in the authors' paper whether and how the dialogue history is utilized. If we compare authors' Fig. 2 with Wang et. al's Fig. 1, it becomes clear that authors' framework description is excessively fragmented. Instead of being a *framework* description focused on isolated fragments of their architecture, an *architecture* description would be much more helpful. Considering how UniCRS is central to their architectural choices, authors could borrow Fig. 2 from Wang et al. and describe their contributions in the context of it.

Better definitions and using the Appendices would greatly benefit this paper (see Reasons to Reject, #1).

Was the parameter study (subsection 4.5) on the validation set? Please specify.

---

> ### Author Rebuttal · Authors · 2023-08-26
>
> 1. The paper lacks clarity in multiple parts. The architecture is presented in a very fragmented way (see Presentation Improvements for more details), and concepts are ill-defined. Importantly, it is never perfectly clear what exactly the "recommender nodes" are --- illustrating a single node and the actual graph would be helpful, even as an Appendix. Consequently, authors use expressions like "this enables direct access to historical information" (lines 539-540), and readers can only assume that "historical information" is defined as past recommender runs --- even though there could be multiple interpretations to "historical information."
>
> **Response:** Thank you for the detailed comments. We apologize for not adequately explaining the relevant concepts, which has constrained the readability of our paper. The recommender-item graph we constructed is a heterogeneous bipartite graph, comprising nodes representing recommenders and items. The edges signify whether the recommender successfully recommended the item in the training set dialogues. If a recommender successfully recommended an item in the training set, an edge connects the recommender node to the item node with an edge weight of 1. The term "historical information" refers to the historical recommender-item recommendation results in the training set. During inference, the prompt can retrieve relevant recommender-item interactions from the graph to provide more effective guidance. We hope this provides a clearer understanding of the graph construction.
>
> 2. The second part of the results, measured in terms of Dist-2, Dist-3, and Dist-4, is not particularly convincing. Given the more dynamic nature of prompt learning, there will be higher prompt variation; therefore, it is highly expected that there will be greater natural language variation at generation. For this reason, Dist-n should not be the measure to differentiate between methods. Instead, the evaluation question should be: How helpful is this increased natural language variation? Authors hint on this question with two qualitative examples on Table 2, but aspects like helpfulness/informativeness should be more quantitatively measured --- for example, by using human judges on a certain sample of responses.
>
> **Response:** Many thanks for your suggestion! We have taken your suggestions into serious consideration. Conversational recommender systems (CRS) aim to recommend high-quality items through natural language conversations. It consists of two subtasks: item recommendation and response generation. In the item recommendation subtask, our goal is to offer accurate recommendations to users. In the response generation subtask, our aim is to assist in the recommendation task with diverse responses. Therefore, the task emphasizes accuracy in item recommendation and seeks maximum diversity in response generation. Following previous CRS works such as UniCRS and KGSF, we adopt Distinct-𝑛 (𝑛=2,3,4) at the word level to evaluate the diversity of the generated responses.
>
> To further verify the effectiveness of our model in terms of the generation quality, we have extended our evaluation metrics by incorporating the BLEU metric alongside the Distinct-𝑛 scores.  We present these additional metrics in Table I, comparing the performance of our proposed DOP model with the best-performing baseline, UniCRS. The experimental results clearly demonstrate our method maintains a strong performance in response generation accuracy, which proves that our DOP successfully captures the personal language habits of the recommender and integrates contextual information through the open-book prompt and is more effective for response generation.
>
> **Table I. Supplement results on the conversation task.**
> |           | Dist-2 | Dist-3 | Dist-4 | BLEU-2 | BLEU-3 | BLEU-4 |
> |:----------|-------:|-------:|-------:|-------:|-------:|-------:|
> | UniCRS    |  0.437 |  0.689 |  0.880 |  0.055 |  0.029 |  0.017 |
> | DOP       |  0.876 |  1.428 |  1.827 |  0.057 |  0.034 |  0.021 |
>
>
> Moreover, according to your suggestion, we have gone a step further by conducting human evaluations to complement the automated metrics. Following UniCRS, we invite three annotators to score the generated responses of our model and the best-performing baseline for 200 utterances from two aspects, namely Fluency and Informativeness. The range of scores is 0 to 2. Fluency measures how likely the generated text is produced by human. Informativeness measures whether the response provides new information and knowledge in addition to the post. The results are shown in the Table II below. As can be seen, the responses we generated are both fluent and rich in information. These results prove that the recommender’s experiences as an open-book store help the model effectively understand the dialogue history, and generate fluent and informative responses.
>
> **Table II. Supplement results of human evaluations.**
> |         | Fluency | Informativeness |
> |---------|-------:|---------------:|
> | UniCRS  |   1.39  |           0.97 |
> | DOP     |   1.41  |           1.04 |
>
>
> 3. In terms of the architecture, is there any use for the dialogue history in addition to the current user utterance?
>
> **Response:** In addition to the current user utterance, the dialogue history is integrated in the form of a graph during the prompt construction process. Initially, we establish a recommender-item historical experience bipartite graph based on whether recommenders successfully recommended specific items in the dialogue history. This graph includes nodes representing recommenders and items. It's worth noting that this graph construction process does not consider the explicit user demands expressed within the dialogue. Then, we employ three strategies to retrieve the historical items associated with the ongoing dialogue from the graph to enhance the contextual understanding.
>
> 4. Unlike Wang et. al (2022), it is unclear in the authors' paper whether and how the dialogue history is utilized. If we compare authors' Fig. 2 with Wang et. al's Fig. 1, it becomes clear that authors' framework description is excessively fragmented. Instead of being a framework description focused on isolated fragments of their architecture, an architecture description would be much more helpful. Considering how UniCRS is central to their architectural choices, authors could borrow Fig. 2 from Wang et al. and describe their contributions in the context of it.
>
> **Response:** Many thanks for your suggestion! We use the dialogue history to construct the recommender-item graph which serves as an open-book knowledge store. During inference, we incorporate the current user utterance to retrieve relevant items from the graph, thereby enriching the present context. We recognize the need for a more cohesive framework description and will take your suggestion to heart. To provide a clearer and more comprehensive understanding of our architecture, we plan to enhance our presentation. We will consider your advice to incorporate Fig. 2 from Wang et al.'s work and illustrate our contributions within that context. Once again, we thank you for your constructive feedback, which will undoubtedly contribute to the clarity and cohesiveness of our paper.
>
> 5.	Was the parameter study (subsection 4.5) on the validation set? Please specify.
>
> **Response:** Yes, the parameter study in subsection 4.5 was conducted on the validation set. Specifically, we trained the parameters on the training set, refined the performance by selecting the optimal parameters on the validation set and saved the best-performing model, and ultimately used this model to evaluate the results on the test data.

---

### Official Review · Reviewer_1wDy · 2023-08-05

**Soundness:** 2

**Excitement:**

2: Mediocre: This paper makes marginal contributions (vs non-contemporaneous work), so I would rather not see it in the conference.

**Paper Topic And Main Contributions:**

This paper is focused on conversational recommender systems and specifically aims to improve prompt-based approaches to conversational recommender systems (prompting pre-trained language models to elicit recommendations and dialog utterances). The authors propose to use prefix-tuning guided by a combination of training corpus utterance retrieval and ranking.

**Reasons To Accept:**

- The DOP/dynamic open book approach makes sense intuitively re: adding additional knowledge outside of the conversation context making use of some form of relevance ranking.
- The authors report statistically significant gains in recommendation tasks for DOP over multiple classes of baselines.
- The authors conduct an ablation study to justify their architecture and modeling choices.

**Reasons To Reject:**

- The paper is positioned in a confusing way. The authors sell this paper as a prompt-based approach to primarily leverage pre-trained language models, but the bulk of the method aims to learn a recommender system that is then used to train the LM prefix module/weights. This amounts to a multi-part recommender system like traditional approaches to conversational recommendation.
- This paper would have benefited from a comparison against natural language prompting (or in-context examples drawn from the same training example [open book] selection methodology to derive the prefix) of a pre-trained language model. This would have better justified the decision to use prefix-tuning, etc. Additionally, the assertion that "discrete prompts' impact on model performance is not very significant" (section 2) needs support, especially in light of the past few years' high success rates in adapting PLMs for various NLP/NLU tasks with discrete prompts.
- The choice of metric (solely distinct-n) for evaluating response generation is a bit confusing. There is no measure of fidelity, fluency, or faithfulness - only diversity? The metrics section is the first part in which engagingness is mentioned in earnest so it seems strange that no other metrics are compared. As such, the assertion that "Your DOP model consistently outperforms other models across all metrics, demonstrating its superiority" reads as disingenuous.
- The emphasis on recommendation (e.g. ablation was only conducted for recommendation subtask, the text generation metrics seeming like an afterthought) implies that EMNLP may not be the right venue for this paper; instead it may be better positioned for CIKM or RecSys?

**Reproducibility:**

3: Could reproduce the results with some difficulty. The settings of parameters are underspecified or subjectively determined; the training/evaluation data are not widely available.

**Reviewer Confidence:**

4: Quite sure. I tried to check the important points carefully. It's unlikely, though conceivable, that I missed something that should affect my ratings.

**Typos Grammar Style And Presentation Improvements:**

Leading diagram (Figure 1) is noisy, with small font. Could clean this up and improve readability.

---

> ### Author Rebuttal · Authors · 2023-08-26
>
> 1.	The paper is positioned in a confusing way. The authors sell this paper as a prompt-based approach to primarily leverage pre-trained language models, but the bulk of the method aims to learn a recommender system that is then used to train the LM prefix module/weights. This amounts to a multi-part recommender system like traditional approaches to conversational recommendation.
>
> **Response:** Thank you for bringing this concern to our attention. We apologize for any confusion caused by our explanation of the architecture, but we respectfully disagree with this comment. Our method indeed follows a prompt-based approach for conversational recommendation task. The rationale behind this approach lies in the ability of prompt to unify the formats of both response generation subtask and item recommendation subtask.
>
> Our primary innovation is centered around the construction of the prompt. We utilize the representation of the recommender as the base prompt, allowing us to maximize the utilization of information from the recommendation scenario. Given the limited context within the current utterance, we propose leveraging the recommender's past successful recommendations as an open-book knowledge store, referencing it during inference. Subsequently, we employ three strategies to retrieve historical item recommendation experiences relevant to the ongoing dialogue, thereby enhancing targeting. This enhancement strengthens the representation of the base prompt, infusing it with information about the labels of potentially recommended items. As a result, the language model's capability for the recommendation task is enhanced. Our approach distinctively aims to integrate the recommendation scenario knowledge into the prompt, thereby enabling effective conversational recommendation.
>
>
> 2.	This paper would have benefited from a comparison against natural language prompting (or in-context examples drawn from the same training example [open book] selection methodology to derive the prefix) of a pre-trained language model. This would have better justified the decision to use prefix-tuning, etc. Additionally, the assertion that "discrete prompts' impact on model performance is not very significant" (section 2) needs support, especially in light of the past few years' high success rates in adapting PLMs for various NLP/NLU tasks with discrete prompts.
>
> **Response:** We thank the reviewer for pointing out this issue and we deeply apologize for the absoluteness of the statement in Section 2, which might have led to misunderstandings. Firstly, UniCRS has integrated natural language prompting for the recommendation task, and we use it as our baseline for comparison. Secondly, we intended to convey that the impact of manually designed discrete prompts can vary significantly. Even substituting a single word within a manually designed prompt can result in substantial variations in performance. It's important to note that without meticulously crafting manual prompts and carefully evaluating the impact of each word on performance, using a simplistic manual prompt might not yield optimal results. We deeply value the guidance provided by the reviewer, and we are committed to revising the statement within our paper to better reflect these nuances.
>
> 3. The choice of metric (solely distinct-n) for evaluating response generation is a bit confusing. There is no measure of fidelity, fluency, or faithfulness - only diversity? The metrics section is the first part in which engagingness is mentioned in earnest so it seems strange that no other metrics are compared. As such, the assertion that "Your DOP model consistently outperforms other models across all metrics, demonstrating its superiority" reads as disingenuous.
>
> **Response:** Thanks for your comments! We have taken your suggestions into serious consideration. Conversational recommender systems (CRS) aim to recommend high-quality items through natural language conversations. It consists of two subtasks: item recommendation and response generation. In the item recommendation subtask, our goal is to offer accurate recommendations to users. In the response generation subtask, our aim is to assist in the recommendation task with diverse responses. Therefore, the task emphasizes accuracy in item recommendation and seeks maximum diversity in response generation. Following previous CRS works such as UniCRS and KGSF, we adopt Distinct-𝑛 (𝑛=2,3,4) at the word level to evaluate the diversity of the generated responses.
>
> To further verify the effectiveness of our model in terms of the generation quality, we have extended our evaluation metrics by incorporating the BLEU metric alongside the Distinct-𝑛 scores.
> We present these additional metrics in Table I, comparing the performance of our proposed DOP model with the best-performing baseline, UniCRS. The experimental results clearly demonstrate our method maintains a strong performance in response generation accuracy, which proves that our DOP successfully captures the personal language habits of the recommender and integrates contextual information through the open-book prompt and is more effective for response generation.
>
> **Table I. Supplement results on the conversation task.**
> |           | Dist-2 | Dist-3 | Dist-4 | BLEU-2 | BLEU-3 | BLEU-4 |
> |:----------|-------:|-------:|-------:|-------:|-------:|-------:|
> | UniCRS    |  0.437 |  0.689 |  0.880 |  0.055 |  0.029 |  0.017 |
> | DOP       |  0.876 |  1.428 |  1.827 |  0.057 |  0.034 |  0.021 |
>
>
> Moreover, according to your suggestion, we have gone a step further by conducting human evaluations to complement the automated metrics. Following UniCRS, we invite three annotators to score the generated responses of our model and the best-performing baseline for 200 utterances from two aspects, namely Fluency and Informativeness. The range of scores is 0 to 2. Fluency measures how likely the generated text is produced by human. Informativeness measures whether the response provides new information and knowledge in addition to the post. The results are shown in the Table II below. As can be seen, the responses we generated are both fluent and rich in information. These results prove that the recommender’s experiences as an open-book store help the model effectively understand the dialogue history, and generate fluent and informative responses.
>
> **Table II. Supplement results of human evaluations.**
> |         | Fluency | Informativeness |
> |---------|-------:|---------------:|
> | UniCRS  |   1.39  |           0.97 |
> | DOP     |   1.41  |           1.04 |
>
>
> 4.	The emphasis on recommendation (e.g. ablation was only conducted for recommendation subtask, the text generation metrics seeming like an afterthought) implies that EMNLP may not be the right venue for this paper; instead it may be better positioned for CIKM or RecSys?
>
> **Response:** Thank you for your suggestion. Conversational recommender systems aim to provide recommendations to users through natural language interactions. The foundation lies in the idea that engaging in discussions with personas like friends, librarians, movie rental store attendants, or fellow cinephiles can enrich people's movie choices. This task inherently embraces both recommendation and dialogue elements, with a focus on enhancing the precision of recommended items and diversifying the response generation process. It presents a comprehensive challenge that encompasses both recommendation and conversation domains.
>
> Moreover, our adoption of a prompt-based architecture seeks to leverage the knowledge of pre-trained language models as much as possible, which aligns with natural language processing tasks. While our paper indeed accentuates the recommendation aspect, we also utilize extracted information as prompt to effectively integrate the item recommendation and response generation subtasks within the context of natural language interactions. This approach significantly contributes to its relevance within the scope of EMNLP. We genuinely appreciate your insights concerning potential publication venues, and we firmly believe that our work aligns with the themes and goals of this conference.
>
> 5. Leading diagram (Figure 1) is noisy, with small font. Could clean this up and improve readability.
>
> **Response:** Thank you for your suggestion. We will continue to refine the figure to enhance its readability.

---

### Official Review · Reviewer_MYtF · 2023-08-05

**Soundness:** 2

**Excitement:**

3: Ambivalent: It has merits (e.g., it reports state-of-the-art results, the idea is nice), but there are key weaknesses (e.g., it describes incremental work), and it can significantly benefit from another round of revision. However, I won't object to accepting it if my co-reviewers champion it.

**Paper Topic And Main Contributions:**

This paper introduces a novel method for conversational recommendation through the utilization of prompts. Experimental results conducted on the ReDial datasets demonstrate the notable advantages of the proposed approach.

**Questions For The Authors:**

Can the authors explain the concept of an "open-book prompt" and the reason for using the term "open-book" to describe it?

Could the authors explain why they chose RoBERTa in section 3.2.1 and selected DialoGPT as the base model for generation? Why were other pretrained language models not considered? Have the authors assessed the performance of other PLMs?

The recall@1 score for recommendation currently stands at approximately 5.7%, indicating a rather modest level of effectiveness. In light of this, is there a practical significance to employing such methods in real-world applications?

**Reasons To Accept:**

New methods having better performance.

**Reasons To Reject:**

The evaluation of the generation performance is questionable. The authors primarily present Dist-2, 3, and 4 scores in Table 1 as indicators of generation quality. Nevertheless, based on my personal experience, the generated sentences may exhibit considerable diversity (reflected in high Dist-n scores), yet still exhibit subpar readability and overall quality. Consequently, relying solely on dist-n scores may prove insufficient. To ensure a comprehensive comparison, it would be neccessary to incorporate additional generation metrics like BLEU alongside dist-n scores.

Beyond automated metrics, it is imperative to complement the evaluation of generation performance with human assessments to substantiate the efficacy of the proposed method.

It would be valuable for the authors to provide supplementary experiments that further underscore the effectiveness of their proposed methods. While the ablation study and case study are informative, there exists a curiosity regarding the nuanced performance variations resulting from the application of the proposed mechanisms. It is important to recognize that enhancing performance in isolation may lack significance, particularly if the resultant recommendations are not practically applicable or realistic.

**Reproducibility:**

3: Could reproduce the results with some difficulty. The settings of parameters are underspecified or subjectively determined; the training/evaluation data are not widely available.

**Reviewer Confidence:**

5: Positive that my evaluation is correct. I read the paper very carefully and I am very familiar with related work.

---

> ### Author Rebuttal · Authors · 2023-08-26
>
> 1. The evaluation of the generation performance is questionable. The authors primarily present Dist-2, 3, and 4 scores in Table 1 as indicators of generation quality. Nevertheless, based on my personal experience, the generated sentences may exhibit considerable diversity (reflected in high Dist-n scores), yet still exhibit subpar readability and overall quality. Consequently, relying solely on dist-n scores may prove insufficient. To ensure a comprehensive comparison, it would be neccessary to incorporate additional generation metrics like BLEU alongside dist-n scores. Beyond automated metrics, it is imperative to complement the evaluation of generation performance with human assessments to substantiate the efficacy of the proposed method.
>
> **Response:** Thanks for your comments! We have taken your suggestions into serious consideration. Conversational recommender systems (CRS) aim to recommend high-quality items through natural language conversations. It consists of two subtasks: item recommendation and response generation. In the item recommendation subtask, our goal is to offer accurate recommendations to users. In the response generation subtask, our aim is to assist in the recommendation task with diverse responses. Therefore, the task emphasizes accuracy in item recommendation and seeks maximum diversity in response generation. Following previous CRS works such as UniCRS and KGSF, we adopt Distinct-𝑛 (𝑛=2,3,4) at the word level to evaluate the diversity of the generated responses.
>
> To further verify the effectiveness of our model in terms of the generation quality, we have extended our evaluation metrics by incorporating the BLEU metric alongside the Distinct-𝑛 scores.
> We present these additional metrics in Table I, comparing the performance of our proposed DOP model with the best-performing baseline, UniCRS. The experimental results clearly demonstrate our method maintains a strong performance in response generation accuracy, which proves that our DOP successfully captures the personal language habits of the recommender and integrates contextual information through the open-book prompt and is more effective for response generation.
>
> **Table I. Supplement results on the conversation task.**
> |           | Dist-2 | Dist-3 | Dist-4 | BLEU-2 | BLEU-3 | BLEU-4 |
> |:----------|-------:|-------:|-------:|-------:|-------:|-------:|
> | UniCRS    |  0.437 |  0.689 |  0.880 |  0.055 |  0.029 |  0.017 |
> | DOP       |  0.876 |  1.428 |  1.827 |  0.057 |  0.034 |  0.021 |
>
>
> Moreover, according to your suggestion, we have gone a step further by conducting human evaluations to complement the automated metrics. Following UniCRS, we invite three annotators to score the generated responses of our model and the best-performing baseline for 200 utterances from two aspects, namely Fluency and Informativeness. The range of scores is 0 to 2. Fluency measures how likely the generated text is produced by human. Informativeness measures whether the response provides new information and knowledge in addition to the post. The results are shown in the Table II below. As can be seen, the responses we generated are both fluent and rich in information. These results prove that the recommender’s experiences as an open-book store help the model effectively understand the dialogue history, and generate fluent and informative responses.
>
> **Table II. Supplement results of human evaluations.**
> |         | Fluency | Informativeness |
> |---------|-------:|---------------:|
> | UniCRS  |   1.39  |           0.97 |
> | DOP     |   1.41  |           1.04 |
>
> 2. Can the authors explain the concept of an "open-book prompt" and the reason for using the term "open-book" to describe it?
>
> **Response:** In the "Closed-book" strategy, the training data is used solely to learn model parameters to maximize probabilities of ground-truth. Then, the model performance is evaluated based on the testing dataset without extra information. This is similar to how humans first acquire knowledge from books and then undergo a closed-book examination to recall the learned content. In fact, we are aware that the difficulty level of open-book exams is lower than that of closed-book exams because one can directly locate and copy relevant content from reference book. The term "open-book prompt" is inspired by the concept of human open-book exams. We believe that copying is simpler than relying on memory. If we can enable retrieval in an open-book scenario, it becomes possible to form prompts by retrieving recommendation scenarios relevant to the current dialogue's semantics from the entire training dataset. This approach enriches the prompt's information, allowing its direct utilization during the inference phase by referring to related knowledge in the training set. Therefore, we use the term "open-book" to indicate that our prompt possesses the ability to access open-book knowledge from the training phase and retrieve explicit information during the inference stage.
>
> 3. Could the authors explain why they chose RoBERTa in section 3.2.1 and selected DialoGPT as the base model for generation? Why were other pretrained language models not considered? Have the authors assessed the performance of other PLMs?
>
> **Response:** UniCRS is the first work to develop a unified CRS in a general prompt learning way. This approach employs RoBERTa for encoding sentence semantics and utilizes DialoGPT for the base PLM since it has been pretrained on a large-scale dialogue corpus. In order to maintain experimental fairness and prevent potential performance improvements resulting from modifications to the pre-trained language model, our experimental setup mirrors that of UniCRS. Both RoBERTa and DialoGPT parameters are kept frozen during model training the same as UniCRS.
>
> While we acknowledge the significance of exploring a wide array of pretrained language models, our decision to align with the UniCRS framework was motivated by the desire to avoid inadvertent performance discrepancies that could arise from altering the underlying language model. Although we haven't explicitly evaluated other pretrained language models in this iteration of our study, we recognize this avenue as a promising area for future investigation.
>
> Furthermore, to demonstrate that our DOP does not incur the high computational cost, we compare it with the baseline UniCRS in terms of the parameter number and running time on two subtasks. From Table III below, we can conclude that our model does not add much cost and is relatively faster than the baseline.
>
> **Table III. Computational Cost Comparison.**
> | Model  | Recommendation    |                   | Conversation      |                    |
> |--------|-------------------|-------------------|-------------------|--------------------|
> |        | Parameter Number  | Runtime per Epoch | Parameter Number  | Runtime per Epoch  |
> | UniCRS | 585M              | 1402.8s           | 585M              | 2501.2s            |
> | DOP    | 531M              | 541.4s            | 531M              | 2132.5s            |
>
> 4.	The recall@1 score for recommendation currently stands at approximately 5.7%, indicating a rather modest level of effectiveness. In light of this, is there a practical significance to employing such methods in real-world applications?
>
> **Response:** With the emergence of social chatbots, conversational recommender systems have gained prominence as a focal point, aiming to provide users with recommendation services through natural language interactions.
>
> Firstly, engaging in conversations with personas like friends, librarians, movie rental store attendants, or fellow cinephiles is an enjoyable experience that brings new perspectives to people's movie choices. Our work not only addresses fundamental conversational recommendation tasks but also incorporates variations in the roles of recommenders. This allows us to offer advice to users from different angles, enhancing the element of surprise. For example, engaging in discussions with a Marvel fan is more likely to result in recommendations for Marvel movies, whereas interactions with a librarian could lead to suggestions for films adapted from books. These distinct roles bring diverse recommendation influences, thereby augmenting diversity and contributing to heightened user satisfaction. That’s the reason why we integrate the recommenders’ historical experiences.
>
> Furthermore, it's important to acknowledge that the effectiveness of existing methods can be constrained by the limited context available in ongoing conversations. Indeed, selecting the most suitable movie for a user from hundreds or thousands of options based solely on the limited conversation is a challenging task. Our approach takes a crucial step toward addressing this limitation. By incorporating role-specific recommender experiences and enabling the retrieval of pertinent information, our method enhances contextual awareness even within the constraints of a brief dialogue. This enhancement, in turn, results in a notable improvement in the performance of Recall@1.
>
> In conclusion, the practical significance of employing such methods in real-world applications lies in their ability to infuse engagement, novelty, and personalization into recommendation services while simultaneously addressing the challenges of limited context. Through the integration of diverse recommenders' roles and experiences, we create a more dynamic and effective recommendation experience for users.

---

### Meta-Review · Area_Chair_vYwo · 2023-09-19

**Recommendation:** 2

**Metareview:**

This paper proposes a novel Dynamic Open-book Prompt approach for conversational recommendation, where the open book stores users' experiences in historical data, and the prompt is dynamically constructed to memorize the user's current utterance and selectively retrieve relevant contexts from the open book. Extensive experimental results on the ReDial dataset demonstrate the significant improvements achieved by the proposed model over the state-of-the-art methods.

All reviewers have pointed out that the evaluation of the generation performance is questionable by using only metrics Dist-n. Although the results of BLEU are added in the rebuttal, its validity should be justified given the multiple acceptable responses that can vary in syntax/lexicon, and some contributions also seem to be overclaimed. Reviewers further suggest using more recent and more powerful PLMs rather than DialoGPT in the experiments. Besides, reviewer MJfa mentioned that the paper lacks clarity in many parts, which need to be fixed.

---

### Meta-Review · Senior_Area_Chairs · 2023-10-05

**Recommendation:** 3

**Metareview:**

meta review

---

### Decision · Program_Chairs · 2023-10-07

**Decision:**

Accept-Findings

**Comment:**

This paper proposes a novel Dynamic Open-book Prompt approach for conversational recommendation, where the open book stores users' experiences in historical data, and the prompt is dynamically constructed to memorize the user's current utterance and selectively retrieve relevant contexts from the open book. Extensive experimental results on the ReDial dataset demonstrate the significant improvements achieved by the proposed model over the state-of-the-art methods.

All reviewers have pointed out that the evaluation of the generation performance is questionable by using only metrics Dist-n. Although the results of BLEU are added in the rebuttal, its validity should be justified given the multiple acceptable responses that can vary in syntax/lexicon, and some contributions also seem to be overclaimed. Reviewers further suggest using more recent and more powerful PLMs rather than DialoGPT in the experiments. Besides, reviewer MJfa mentioned that the paper lacks clarity in many parts, which need to be fixed.|meta review